

# SENSITIVITY OF STOMATAL CONDUCTANCE TO SOIL MOISTURE:

# IMPLICATIONS FOR TROPOSPHERIC OZONE

Alessandro Anav[1*], Chiara Proietti[1], Laurent Menut[2], Stefano Carnicelli[3], Alessandra De Marco[4],

Elena Paoletti[1]

[1]*Institute of Sustainable Plant Protection, National Research Council , Sesto Fiorentino, Italy.*

[2]*Laboratoire de Meteorologie Dynamique, LMD/IPSL, École Polytechnique, Palaiseau, France.*

[3]*Earth Sciences Department, University of Florence, Florence, Italy*

[4]*Italian National Agency for New Technologies, Energy and the Environment (ENEA), C.R. Casaccia,*

*S. Maria di Galeria, Italy.*

*Correspondence to*: Alessandro Anav (alessandro.anav@ipsp.cnr.it)

## ABSTRACT

Soil moisture and water stress play a pivotal role in regulating stomatal behaviour of plants; however, in the last decade, the role of water availability was often neglected in atmospheric chemistry modelling studies as well as in integrated risk assessments, despite through stomata plants remove a large amount of atmospheric compounds from the lower troposphere.

The main aim of this study is to evaluate the effect of soil water limitation on stomatal conductance and assess the resulting changes in atmospheric chemistry testing various hypotheses of water uptake by plants in the rooting zone; following the main assumption that roots maximize water uptake, i.e. they adsorb water at different soil depths depending on the water availability, we improve the dry deposition scheme within the chemistry transport model CHIMERE.

Results highlight how dry deposition significantly declines when soil moisture is used to regulate the stomatal opening, mainly in the semi-arid environments: in particular, over Europe the amount of ozone removed by dry deposition in one year without considering any soil water limitation to stomatal



conductance is about 8.5 TgO$_3$, while using a dynamic layer that ensures plants to maximize the water
uptake from soil, we found a reduction of about 10% in the amount of ozone removed by dry
deposition (~7.7 TgO$_3$). Despite dry deposition occurs from top of canopy to ground level, it affects the
concentration of gases remaining into the lower atmosphere with a significant impact on ozone
concentration (up to 4 ppb) extending from the surface to the upper troposphere (up to 650 hPa).
Our results shed light on the importance of improving the parameterizations of processes occurring at
plant level (i.e. from the soil to the canopy) as they have significant implications on concentration of
gases in the lower troposphere.

## 1.    Introduction

Plant-level water cycling and exchange of air pollutants between atmosphere and vegetation are
intimately coupled (Eamus, 2003; Domec et al., 2010), thus any factor affecting root water absorption
by plants is expected to impact the concentration of gases in the lower troposphere by changing
deposition rates. In fact, atmospheric gases, including air pollutants, are primarily removed from the
troposphere by dry deposition to the Earth's surface (Hardacre et al., 2015; Monks et al., 2015). A
major part of dry deposition to vegetation is regulated by stomata opening which strongly depends on
the amount of water available in the soil (Büker et al., 2012). Therefore a proper quantification of soil
water content as well as a proper understanding of stomatal response to soil moisture are required for
correctly quantifying the concentration of gases in the atmosphere, particularly in water-limited
ecosystems (dry and semidry environments) which cover 41% of Earth's land surface (Reynolds et al.,

49  2007).

Among common air gasses, ozone (O$_3$) plays a pivotal role in the Earth system: in fact, it affects
climate with a direct radiative forcing of 0.2-0.6 W m$^{-2}$ (Shindell et al., 2009, 2012; Ainsworth et al.,
2012; Myhre et al., 2013) and the ecosystems, causing a reduction of carbon assimilation by vegetation
(Wittig et al., 2009) that accelerates the rate of rise in CO$_2$ concentrations with indirect implications for
climate change (Sitch et al., 2007). In addition, O$_3$ accelerates leaf senescence (Gielen et al., 2007),
changes plants susceptibility to abiotic and biotic stress factors (Karnosky et al., 2002) and makes
sluggish or impaired response of stomata to environmental stimuli (Hoshika et al., 2015).
At European level, the model currently parameterized for European vegetation and developed to
estimate surface O$_3$ fluxes is the DO$_3$SE (Deposition of O$_3$ and Stomatal Exchange) model (Emberson
et al., 2000); it is widely used embedded within chemistry transport models (CTMs) (Tuovinen et al.,
2004; Simpson et al., 2007,2012; Menut et al., 2013) to estimate dry deposition rates as well as stand-



alone for O$_3$ risk assessment (Emberson et al., 2007; Tuovinen et al., 2009; Klingberg et al., 2014;
Anav et al., 2016; Sicard et al., 2016; Karlsson et al., 2017). The DO$_3$SE model is based on the
multiplicative Jarvis' algorithm for calculation of stomatal conductance (Jarvis 1976), which integrates
the effects of multiple climatic factors, vegetation characteristics and local features (Emberson et al.,
2000). The leaf-level stomatal conductance is estimated considering the variation in the maximum
stomatal conductance (g$_{max}$) with photosynthetic photon flux density, surface air temperature, and
vapour pressure deficit. However, this original formulation of the DO$_3$SE model presented a main
limitation (Simpson et al., 2007; Tuovinen et al., 2009; Mills et al., 2011): for both forests and crops
the model did not take into account the limitation due to soil water content. This approach ensured that
stomatal fluxes were maximized, corresponding to conditions expected for irrigated areas (Simpson et
al., 2007), but, in semi-arid environments, like the Mediterranean basin, the amount of atmospheric
gases entering the leaves might be compromised by the exclusion of the influence of drought on
stomatal conductance (Tuovinen et al., 2009; Mills et al., 2011; Büker et al., 2012; Anav et al., 2016;
De Marco et al., 2016). Following this assumption, the role of soil moisture on stomatal O$_3$ fluxes has
been often neglected in risk assessment studies because soil water is very difficult to model accurately
in large-scale models, as it depends on parameters (such as soil texture, vegetation characteristics and
rooting depth) that are not easily available in the frame of large scale models (Simpson et al., 2007;
Büker et al., 2012; Simpson et al., 2012).
However, in the last decade the importance of soil water stress on vegetation has been well
demonstrated in several studies reporting a large reduction in the amount of air gases up-taken from the
atmosphere during heat waves or drought years (e.g. Ciais et al., 2005; Granier et al., 2007; Reichstein
et al., 2007) with species responding in different ways to scarce water availability, depending on eco-
hydrological properties (Granier et al., 1996; Pataki et al., 2000; Pataki and Oren, 2003) and drought
avoidance and tolerance strategies (Martinez-Ferri et al., 2000; Bolte et al., 2007). For instance,
drought-avoiding species (e.g. *Pinus spp.*) prevent damage by an early stomatal closure that leads to a
sharp carbon assimilation inhibition, whereas drought-tolerant species (e.g. *Quercus spp.*) exhibit a
simultaneous decrease in stomatal conductance and water potential (Guehl et al,. 1991, Picon et al.,
1996) that does not significantly limit carbon assimilation. Nevertheless, both strategies have severe
implications on the concentration of gases in the lower troposphere.
Moreover, it is important to take into account that soil drying does not occur at the same rate at
different depths, and the drying rate is more pronounced in the superficial soil layers than in the deeper
ones. Overall, deep-rooted forest systems take up water from deep to shallow soil horizons (Aranda et





al., 2012). In contrast, shallow-rooted grass normally adsorbs available soil water from top−middle
soil, while shrubs can take up soil water adaptively from top to deep soil layers, with increased use of
top-soil water under non-drought stress and a tendency of using water from deeper soil under drought
stress (Wu et al., 2017). Thus, plants able to develop a deeper root system usually are more tolerant to
low water availability than plants with a more superficial root system (Canadell et al., 1996). Jackson
et al. (2000) showed that differences in rooting depth patterns vary between world's major plant
biomes, with plants of xeric environments having deeper root-depth distributions than plants in more
humid environments. In contrast, Schenk and Jackson (2002) found that maximum rooting depths tend
to be shallowest in arid regions and  deepest  in  sub-humid  regions.
Consequently, the role of root systems is fundamental in stomatal conductance regulation and thus in
atmospheric chemistry modeling. For these reasons, recently the $DO_3SE$ model has been improved to
account for the soil moisture limitation to stomatal conductance (Büker et al., 2012; Simpson et al.,

105  2012).

Chemistry transport models are widely used to estimate the concentration of gases in atmosphere at
both regional and global scale; in these models the concentration of a given gas-species is mainly
regulated and parameterized by three different processes: atmospheric transport, chemical
production/destruction and losses to surface by dry deposition (Monks et al., 2015). Within these
models, the dry deposition is generally simulated through an electrical resistance analogy (Wesely
1989; Monk et al., 2015), namely the transport of material to the surface is assumed to be controlled by
three different resistances: the aerodynamic resistance ($R_a$), the quasi-laminar layer resistance ($R_b$), and
the surface resistance ($R_c$). The surface resistance is regulated by the stomatal uptake, which relies on
stomatal conductance, as well as external plant surfaces like the soil underlying the vegetation.
In this study, we improve the dry deposition scheme within the chemistry transport model CHIMERE
considering the effect of soil water limitation to stomatal conductance. Our main aim was to perform
several different simulations testing various hypotheses of water uptake by plants at different soil
depths in the rooting zone, based on the main assumption that roots maximize water uptake to fulfill
resource requirements adsorbing water at different depths depending on the water availability. Finally
we show and discuss the resulting effects on $O_3$ dry deposition and concentration, in order to stress the
need of a proper parameterization of root-depth soil moisture when evaluating the stomatal feedbacks
on the atmosphere and for a thorough $O_3$ risk assessment.




## 2. Methodology

### 2.1. The multi-model framework

We use a multi-model system to reproduce the meteorological conditions and the concentration of gases in the troposphere; this framework is composed by the WRF (Weather Research and Forecast Model) regional meteorological model and the CHIMERE chemistry-transport model.

In this study, in order to have a large latitudinal gradient and assess the role of soil moisture across different climate zones, we selected a domain extending over all Europe (except Iceland). For both WRF and CHIMERE we performed a simulation for the whole year 2011, with a spin up of 2 months to initialize all the fields.

### 2.1.1. The meteorological model WRF

Meteorological variables are simulated with the WRF regional model (v 3.6); it is a limited-area, non-hydrostatic, terrain-following eta-coordinate mesoscale model (Skamarock et al., 2008) widely used worldwide for climate studies. In our configuration, the model domain is projected on a regular latitude-longitude grid with a spatial resolution of 16 km and with 30 vertical levels extending from land surface to 50 hPa. The initial and boundary meteorological conditions required to run the WRF model are provided by the European Centre for Medium-range Weather Forecast (ECMWF) analyses with a horizontal resolution of 0.7° every 6 hours (Dee et al., 2011).

The exchange of heat, water and momentum between soil-vegetation and atmosphere is calculated using the Noah land surface model (Chen and Dudhia, 2001); in our configuration the soil has a vertical profile with a total depth of 2 m below the surface and it is partitioned into four layers with thicknesses of 10, 30, 60, and 100 cm (giving a total of 2 m). The root zone is fixed at 100 cm (i.e. including the top three soil layers). Thus, the lower 100 cm of soil layer acts as a reservoir with gravity drainage at the bottom (Al-Shrafany et al., 2013).

For each soil layer Noah calculates the volumetric soil water content (θ) from the mass conservation law and the diffusivity form of Richards' equation (Chen and Dudhia, 2001):

$$\frac{\partial \theta}{\partial t} = \frac{\partial \theta}{\partial z}\left(D\frac{\partial \theta}{\partial z}\right) + \frac{\partial K}{\partial z} + F_\theta \qquad (1)$$

where D is the soil water diffusivity, K is the hydraulic conductivity, $F_\theta$ represents additional sinks and sources of water (i.e., precipitation, evaporation and runoff), t is time and z is the soil layer depth



(Chen and Dudhia, 2001; Al-Shrafany et al., 2013; Greve et al., 2013). Integrating Eq. (1) over four soil layers and expanding $F_\theta$, we can calculate the volumetric soil water content for each soil layer (Chen and Dudhia, 2001; Al-Shrafany et al., 2013):

$$d_{z1} \frac{\partial \theta_1}{\partial t} = -D\left(\frac{\partial \theta}{\partial z}\right)_{z1} - K_{z1} + P_d - R - E_{dir} - E_{t1} \qquad (2)$$

$$d_{z2} \frac{\partial \theta_2}{\partial t} = D\left(\frac{\partial \theta}{\partial z}\right)_{z1} - D\left(\frac{\partial \theta}{\partial z}\right)_{z2} + K_{z1} - K_{z2} - E_{t2} \qquad (3)$$

$$d_{z3} \frac{\partial \theta_3}{\partial t} = D\left(\frac{\partial \theta}{\partial z}\right)_{z2} - D\left(\frac{\partial \theta}{\partial z}\right)_{z3} + K_{z2} - K_{z3} - E_{t3} \qquad (4)$$

$$d_{z4} \frac{\partial \theta_4}{\partial t} = D\left(\frac{\partial \theta}{\partial z}\right)_{z3} + K_{z3} - K_{z4} \qquad (5)$$

where, $d_{zi}$ is the thickness of the $i$th soil layer, $P_d$ is the precipitation not intercepted by the canopy, $E_{ti}$ represents the canopy transpiration taken by the canopy root in the $i$th layer within the root zone, $E_{dir}$ is the direct evaporation from the top surface soil layer, and R is the surface runoff, calculated using the Simple Water Balance (SWB) model (Schaake et al., 1996). In the deeper soil layer (i.e. 4$^{th}$) the hydraulic diffusivity is assumed to be zero, so that the soil water flux is due only to the gravitational percolation term $K_{z4}$ (i.e. drainage). A full and detailed description of the above mentioned parameterizations used by the Noah scheme can be found in Chen and Dudhia (2001).

For the definition of vegetation and land cover WRF uses the United States Geological Survey (USGS) land cover dataset, which has a resolution of 1km with 24 categories (Loveland et al., 2000; Hibbard et al., 2010; Sertel et al., 2010); this land cover dataset is derived from the 1 km satellite Advanced Very High Resolution Radiometer (AVHRR) data. In addition to land cover, WRF defines 12 soil types and four non-soil types, including organic material, water, bedrock, and ice. Soil types are classified based on the percentage of sand, silt, and clay in the soil (Dy and Fung, 2016); for each soil type, WRF has a default soil parameter table that generalizes the hydraulic and thermal properties of the soil. Soil texture data are derived from the 5-minute Food and Agriculture Organization's (FAO) 16 categories soil types.

One useful capability of WRF is its flexibility in choosing different dynamical and physical schemes; **Table 1** lists the main options used in this study for physical schemes.





183          **Table 1.** WRF 3.6 physical configurations used in the model simulations.

| Process | Configuration | Reference |
|---|---|---|
| Microphysics | Single Moment-3 class (mp_physics = 3)[*] | Hong *et al.* (2004) |
| Cumulus Parameterization | Kain–Fritsch (cu_physics = 1)[*] | Kain (2004) |
| Shortwave Radiation | RRTM (ra_sw_physics = 1)[*] | Mlawer *et al.* (1997) |
| Longwave Radiation | RRTM (ra_lw_physics = 1)[*] | Mlawer *et al.* (1997) |
| Land-surface | Noah land model (sf_surface_physics = 2)[*] | Chen and Dudhia (2001) |
| Planetary Boundary Layer | YSU (bl_pbl_physics = 1)[*] | Hong *et al.* (2006) |

[*]A complete description of parameterizations and model's flags is given in the WRF 3 user guide

185          (http://www2.mmm.ucar.edu/wrf/users/docs/user_guide_V3.6/ARWUsersGuideV3.6.1.pdf)


### 2.1.2.  The chemistry-transport model CHIMERE

The chemistry transport model used in this study is CHIMERE (v2014b), an Eulerian model developed
to simulate gas-phase chemistry, aerosol formation, transport and deposition at regional scale (Menut
et al., 2013).
The gas-phase chemical mechanism used by CHIMERE is MELCHIOR2 (Lattuati, 1997), which
consists of a simplified version (40 chemical species, 120 reactions) of the full chemical mechanism
MELCHIOR; this latter describes more than 300 reactions of 80 species. Photolysis rates are explicitly
calculated using the FastJ radiation module (Wild et al., 2000), as described by Mailler et al. (2016;
2017). External meteorological forcing required by CHIMERE to calculate the atmospheric
concentrations of gas-phase and aerosol species are directly provided by the WRF simulation. In
addition, to accurately reproduce the gas-phase chemistry, emissions must be provided every hour for
the specific species of the chemical mechanism. For studies over Europe, the EMEP inventory
(Vestreng et al., 2009) is usually used for anthropogenic emissions of $NO_x$, CO, $SO_2$, $PM_{2.5}$ and $PM_{10}$.
Biogenic emissions of six species (isoprene, α-pinene, β-pinene, limonene, ocimene, and NO) are
calculated through the MEGAN model (Guenther et al., 2006). This model parameterizes the bulk
effect of changing environmental conditions using three time-dependent input variables: surface air
temperature, radiation and foliage density (i.e. LAI). In the standard version of CHIMERE, LAI
database is given as a monthly mean product derived from MODIS observations, referred to base year
2000 (Menut et al., 2013). However, as climate change leads to a widespread greening of Earth surface
(Zhu et al., 2016), a mean climatological LAI referred to year 2000 could not be adequate to correctly
simulate biogenic emissions during our simulation (year 2011). Thus, here we replaced the original
LAI data with mean monthly GIMMS-LAI3g data (Zhu et al., 2013) for the year 2011.





Boundary conditions are provided as a monthly climatology of the LMDz-INCA global chemistry-
transport model (Hauglustaine et al., 2004; Folberth et al., 2006) for gaseous species and the GOCART
model (Ginoux et al., 2001) for aerosol species. More details regarding the parameterizations of the
above mentioned processes are described in Menut et al. (2013).

**2.1.3.   Dry deposition: the DO₃SE model**
The leaf-level stomatal conductance is estimated by CHIMERE using the DO₃SE model (Emberson et
al., 2000). As already introduced above, this model integrates the effects of multiple climatic factors,
vegetation characteristics and local features through some limiting functions (e.g. Emberson et al.,
2000). The limiting functions consider the variation in the maximum stomatal conductance ($g_{max}$) with
photosynthetic photon flux density ($f_{light}$), surface air temperature ($f_{temp}$) and vapour pressure deficit
($f_{VPD}$) (Mills et al., 2011; Büker et al., 2012); they vary between 0 and 1, with 1 meaning no limitation
to stomatal conductance (e.g. Emberson et al., 2000; Mills et al., 2011). In addition, the DO₃SE model
requires another function describing the phenology of vegetation ($f_{phen}$); this function is used to
compute the duration of growing season during which plants can uptake gases from atmosphere (Anav
et al., 2017).
Here, we improve the DO₃SE scheme within CHIMERE considering also the soil water content (SWC)
limitation to stomatal conductance; the soil-water limitation function is defined as:

$$f_{SWC} = \min\left[1, \max\left(f_{\min}, \frac{SWC - WP}{FC - WP}\right)\right] \qquad (6)$$

where WP and FC are the soil water content at wilting point and at field capacity, respectively; these
two parameters are constant and depend on the soil type. Given the above-mentioned limiting
functions, the stomatal conductance is computed as following:

$$g_{sto} = g_{\max} * f_{phen} * f_{light} * \max(f_{\min}, f_{temp} * f_{VPD} * f_{SWC}) \qquad (7)$$


where $g_{max}$ is the maximum stomatal conductance of a plant species to O₃ and $f_{min}$ is the minimum
stomatal conductance expressed as a fraction of $g_{max}$ (Emberson et al., 2000).



Meteorological fields required by the DO$_3$SE model, such as 2m air temperature, relative humidity,
short wave radiation and soil moisture, are directly provided by WRF. As already discussed above,
WRF computes soil moisture over four soil layers of different thicknesses. For the integrated risk
assessment studies, some authors make use of 1m soil layer to compute the stomatal O$_3$ flux and dry-
deposition (e.g. Simpson et al., 2012), while other authors use a shallower soil moisture layer (e.g. De
Marco et al., 2016) as most of the absorbing fine roots concentrate in the top soil layer (Jackson et al.,
1996; Vinceti et al., 1998). Here we perform five different simulations testing various hypotheses: 1)
no soil moisture limitation to stomatal conductance (henceforth *NO_SWC*), 2) soil moisture from first
soil layer (i.e. 0-10 cm depth, henceforth *SWC_10cm*), 3) soil moisture from middle soil (i.e., 10-40 cm
depth, henceforth *SWC_40cm*), 4) soil moisture from the deeper soil layer of rooting zone (i.e., 0.4-1 m
depth, henceforth *SWC_1m*) and 5) a dynamic layer (henceforth *SWC_DYN*) supporting the hypothesis
that plants adsorb water at the depth with the higher water content availability.
As the original version of CHIMERE does not account for any limitation of soil moisture to stomatal
conductance, in the following analysis we use the simulation *NO_SWC* as reference; thus we show and
discuss models' changes with respect to this original configuration (Menut et al., 2013).

### 2.2.    Measurement data and statistical analysis

In order to assess how the new parameterization of dry deposition changes the ability of CHIMERE to
reproduce the spatial distribution of surface O$_3$ concentration, we compare the simulated data at
surface level against in-situ measurements. Station data were obtained from the European air quality
database (AirBase) and maintained by the European Environment Agency (EEA)
(http://acm.eionet.europa.eu/databases/airbase/).
For the validation of O$_3$ bias, computed comparing hourly simulated O$_3$ concentrations with AirBase
data, we use the root-mean-square error (RMSE), while to assess the agreement in the phase (i.e.
hourly cycle) we use the correlation coefficient.
Considering the soil moisture, we retrieve precipitation data over four forested eddy covariance sites
belonging to the European flux network (http://www.europe-fluxdata.eu); in fact, a good representation
of precipitation simulated by the model is mandatory to correctly reproduce the dynamics of water in
the soil. The choice of these specific sites is due to the multiple requirements of having full year data
coverage with different climatic zones. Specifically, the sites cover a continental climate typical of
central Europe, where soil moisture barely limits the stomatal opening, and Mediterranean sites
characterized by scarce water availability during summer months. Unfortunately, despite soil moisture



is measured in these sites, the depth of measurements is not consistent with model's layers and also it
does not reach the same depth of the model making thus awkward any comparison of the vertical
distribution of water in the soil.

## 3.    Results
### 3.1.    Seasonal changes in soil water content

**Figure 1** shows the seasonal variation of simulated soil water content at four different locations; in
order to assess the reliability of vertical soil moisture profiles we also evaluate models skills in
capturing precipitation events by comparing the simulated precipitation with data collected over the
four measurements stations.
The first site, Leinefelde in Germany, is characterized by a temperate/continental climate with mean
annual precipitation ranging between 700 and 750 mm, covered by a beech forest (*Fagus sylvatica*).
Overall, compared to in-situ observations, WRF well reproduces both the rainfall events and their
intensity (**Figure 1a**). Considering the soil moisture, at the beginning of the year, the soil is at field
capacity, and rapidly becomes saturated down to 40 cm, while below 1m depth from end of January to
mid-April the soil is close to the field capacity. After mid-April, soil remarkably dries out at all depths,
and water content oscillates between 0.28 and 0.36 $m^3 \cdot m^{-3}$ until October, when decreasing evaporative
demand and weak rain events caused a transient partial recovery around 0.33 $m^3 \cdot m^{-3}$. Then, the new
rainfall events at the end of November lead to rising soil water content above the field capacity until
the end of the year (**Figure 1a**).
The second temperate site, covered by a spruce forest (*Picea* abies), is Oberbärenburg in Germany; it is
characterized by a mean annual precipitation of about 1000 mm. Noteworthy, WRF captures most of
the rainfall events, despite it slightly underestimates their intensity during the period May-August.
Here, in the rooting zone, the soil is constantly above the field capacity and near saturation until mid-
March; then it rapidly drains, and soil water content remains in the range 0.24–0.26 $m^3 \cdot m^{-3}$, with short-
term increases following precipitation events, until December, when it increased to above 0.28 $m^3 \cdot m^{-3}$
(**Figure 1b**).
In Collelongo, a *Fagus sylvatica* mountain forest site in central Italy, the mean annual precipitation is
about 1200 mm. From the beginning of the year to the end of June, the soil water content is above 0.3
$m^3 \cdot m^{-3}$, with short term increases above field capacity from 10 cm to 1m and a stable content above
field capacity below 1m depth; then, in July, soil moisture progressively decreases to about 0.20 $m^3 \cdot m^{-}$
$^3$ with a short term rainfall resupply at the end of the month. From August to November, because of



high evapotranspiration rates and weak precipitation events, soil moisture sharply drops to 0.15 $m^3 \cdot m^{-3}$
or less, and, at 1m depth, it appears to have been constantly at wilting point from end of September to
early November. Finally, in December, soil moisture rapidly increases in the upper layers, reaching
near saturation in late December, but remains low around 1m depth until the end of the year (**Figure**
**1c**).
The fourth station is San Rossore, a Mediterranean *Pinus spp.* forest located on the coastal region of
central Italy and characterized by a mean annual precipitation of 920 mm. Here the pattern is
substantially similar to Collelongo: soil water content is lower in spring, when rainfall infiltrates faster
and deeper and less water is retained; the fall drought at 1m depth is less pronounced and of shorter
duration, but water recharge towards the end of the year was again slower (**Figure 1d**).
Overall, these results suggest that soil water availability was higher from April to September for the
two Central European sites, where soil water content remained above 50% of total available water
capacity. In the Mediterranean sites, water availability declined from spring onwards, but remained
above 40% total available water capacity until late August, while effective drought conditions occurred
in October.

**3.2.    Changes in O₃ dry deposition**
The inclusion of soil water limitation in the stomatal conductance parameterization affects, at first, the
surface resistance, that, in turn, affects the dry deposition velocity and thus the amount of air pollutants
removed from the surface layer by dry deposition (Seinfeld and Pandis, 2016; Hardacre et al., 2015;
Monks et al., 2015). **Figure 2** shows the mean percentage of change in O₃ dry deposition during the
periods April-May-June (AMJ) and July-August-September (JAS) between the reference simulation
(i.e. *NO_SWC*) and the simulations that take into account the soil moisture limitation to stomatal
conductance. Clearly, as the inclusion of soil water stress leads to a reduction of stomatal conductance,
the amount of O₃ removed by dry deposition is always larger in the *NO_SWC* simulation than in the
other simulations; this explains the negative pattern in the percentage of change in O₃ dry deposition in
both the analyzed seasons. Looking at the spatial pattern (**Figure 2**), we find the weaker differences in
Norway, where soil moisture is barely limiting the stomatal conductance, while the larger differences
occur in the Mediterranean basin (i.e. Spain, South France, Italy, Greece and Turkey). In fact, in these
semi-arid regions the soil dries out quickly, especially during summer (**Figure 1**), and plants close
their stomata during the warmer hours of the day to prevent water loss, leading to a smaller amount of
O₃ entering the leaves and thus removed by vegetation. This process is well displayed during JAS in



the *SWC_10cm* simulation and to a lesser extent in the *SWC_40cm*, *SWC_1m* and *SWC_DYN*
simulations: specifically, in Southern Europe the upper soil layer (i.e. 10 cm) dries out faster than the
deeper ones during the warm and dry season, consequently, in the *SWC_10cm* simulation we find the
stronger limitation of soil moisture to stomatal conductance and the highest reduction in $O_3$ dry
deposition. In the other simulations we use a deeper rooting zone where plants can uptake water from
the soil; during summer these layers are generally moister than the shallow layer, thus the stomatal
conductance will be less limited by soil moisture and the vegetation removes a larger amount of $O_3$. In
addition to the larger stomatal conductance, during JAS, compared to AMJ, the higher leaf area index
(LAI) increases the surface resistance and thus the amount of $O_3$ removed from the surface layer; this
explains the larger $O_3$ dry deposition values found during summer. Overall, during the whole year the
amount of $O_3$ removed by dry deposition (sum of stomatal and non-stomatal deposition) integrated
over the only land points of domain is 8.568 $TgO_3$ in the *NO_SWC* simulation, 7.576 $TgO_3$ (-11.8%) in
the *SWC_10cm*, 7.618 $TgO_3$ (-11.1%) in the *SWC_40cm*, 7.617 $TgO_3$ (-11.1%) in the *SWC_1m,* and
7.693 $TgO_3$ (-10.2%) in the *SWC_DYN.*

**3.3.    Changes in $O_3$ concentration**
As plants uptake atmospheric gases when stomata are open (Cieslik et al., 2009), changes in stomatal
behavior, and thus in dry deposition velocity, affect, in turn, the concentration of compounds
remaining in the lower atmosphere; **Figure 3** shows the mean percentage of change in $O_3$
concentration in the lowest model layer (20-25 meters in our case) between the reference simulation
(i.e. *NO_SWC*) and the other simulations. Unlike **Figure 2**, where we found a systematic negative
percentage of change in the amount of $O_3$ removed by dry deposition, **Figure 3** shows a systematic
positive percentage of change, i.e. a higher concentration of $O_3$ remaining in the atmosphere in the
simulations where soil moisture limits the stomatal conductance. In addition, the higher (i.e. more
negative) is the percentage of change of $O_3$ removed by deposition, the more is the concentration of $O_3$
remaining in the air: **Figure 3** clearly shows how the larger differences in surface $O_3$ concentration are
found during summer (JAS) in the *SWC_10cm* simulation, i.e. the experiment where soil moisture
plays the strongest limitation to stomatal conductance.
Similarly, the vertical mixing in surface layers, largely driven by wind and its interaction with
frictional drag at the surface (Monks et al., 2015), propagates the changes in $O_3$ concentration from the
surface layer to upper layers. **Figure 4** shows the $O_3$ anomaly between the reference simulation and the
simulations with soil water limitation, averaged over the plant growing season, i.e. April-September



(Anav et al., 2017); here we show only grid points with a significant change in $O_3$ concentration (t-test,
95% confidence), while we mask out points where the anomaly is not significant. The larger anomaly
in $O_3$ concentration (up to 4 ppb) is found in the whole Mediterranean basin for the *SWC_10cm*
simulation; interestingly, the anomaly is significant in almost all the grid points except Ireland and
Scotland, which are characterized by high soil moisture levels even during summer, and up to 800 hPa
where we find an $O_3$ anomaly larger than 1 ppb.

**3.4.    Changes in the model performances**
As discussed above, the inclusion of soil water limitation to stomatal conductance leads to increased
$O_3$ concentration due to the reduced dry deposition rates; this clearly affects the model performances in
reproducing both the phase and amplitude of hourly $O_3$ concentration. Therefore, here we validate the
simulated $O_3$ against AirBase measurements.
**Figure 5** (upper panels) shows how the inclusion of the new parameterization leads to an increase of
model-data misfit during the temporal period April-September, being the percentage of change in
RMSE positive in all the ground stations. Overall, the mean RMSE (average over all the stations)
computed comparing hourly data is 17.8 ppb for the *NO_SWC* simulation, 19.5 ppb in the *SWC_10cm*
and *SWC_40cm*, and 19 ppb in the *SWC_1m* and *SWC_DYN* simulations.
Conversely, the new parameterization improves the model skills in reproducing the observed hourly
cycle (**Figure 5**, lower panels), being the percentage of change in correlation coefficient positive in all
the stations. Overall, the mean correlation computed from hourly data is 0.6 for the *NO_SWC*
simulation, 0.62 in the *SWC_10cm* and 0.64 in the *SWC_40cm*, *SWC_1m* and *SWC_DYN* simulations.

**4.    Summary and conclusion**
In this study, we incorporated the soil moisture limitation into the dry deposition parameterization of
CHIMERE model and tested different hypotheses of water uptake by roots. Model simulations with the
improved parameterization indicate that $O_3$ dry deposition significantly declines when soil moisture
regulates the stomatal opening, particularly in Southern Europe where soil is close to the wilting point
during the dry summer. This mechanism, occurring within the soil, in turn, affects the concentration of
gases remaining into the lower atmosphere and, considering the vertical mixing in the boundary layer
and the long-lived species such as $O_3$, has an impact on $O_3$ concentration extending from the plants
canopy to the upper troposphere and decreasing with height; the influence on $O_3$ concentration then
quickly vanishes above the boundary layer, becoming no more significant above 650 hPa.





The analysis of simulated soil moisture suggests that actual water availability from April to September,
even in the Mediterranean sites, is higher than conventionally assumed; according to Allen et al.
(1998) and Martínez-Fernández et al. (2015), soil water content values corresponding to 40-50% of
total available water (TAW, FC-WP) often correspond to low stress conditions for cultivated plants. As
the stress threshold lowers with rooting depth (Allen et al 1998), it appears likely that the effect of
water deficit on forest vegetation is limited in these conditions. As in the modified DO$_3$SE model the
effect of soil water content on stomatal aperture is modeled as a linear function of SWC-WP (eq. 6), it
is possible that the actual reduction in stomatal conductance is overestimated for SWC values above
40-50% of TAW, i.e. the most common condition predicted by WRF in the April–September period
over the analyzed sites.
With the modified parameterization, CHIMERE shows increased bias in the prediction of surface
hourly O$_3$ concentrations across Europe with improved representation of the phase of the hourly cycle.
Therefore the new parameterization increases the well-known systematic overestimation of O$_3$
concentrations (e.g. Anav et al., 2016), which derives from initial and lateral boundary conditions
provided by the global chemistry-transport model LMDz-INCA that overestimate the observed
background concentrations (Terrenoire et al., 2015) as well as from bias in anthropogenic and biogenic
emissions.
It should also be noted that the model comparison to satellite retrievals is not obvious in this study: in
fact, here we mainly focus on O$_3$ changes in the boundary layer and lower troposphere, which
correspond to the part of the atmosphere where satellite data are not robust: as shown by Boynard et al.
(2016), the O$_3$ vertical profiles inversions begin to be efficient in the upper troposphere and in the
stratosphere, where our changes become to be negligible. Therefore, it would be largely uncertain to
extract the signal close to the surface and assess how much our different hypotheses improved the total
O$_3$ column. Similarly, the comparison with vertical soundings would display the simulated vertical
profiles very close each other.
Nevertheless our results can be used to improve the representation of soil moisture stress on vegetation
within chemistry transport models and to better describe the biogeochemical and biophysical feedbacks
between the complex soil-plant-atmosphere system in response to a changing climate toward warmer
and drier conditions. As the soil water uptake is mainly related to different rooting systems (Wu et al.,
2017), chemistry models would benefit from the inclusion of species-specific parameterizations which
ensure a water uptake depending on species-specific eco-hydrological properties. In general, plants in
water-limited regions can adapt to dry environments by accessing ground water (Craine et al., 2013)



based on the depth and density of the root system (Wu et al., 2017), while deep-rooted forests can take
up available water from deep soil during extreme drought events (Schwinning et al., 2005; Teuling et
al., 2010). Although some of these processes are already well resolved within land surface models used
by climate models, a better description of different rooting systems within the dry deposition schemes
might have significant implication for stomatal regulation and thus atmospheric chemistry. We also
believe that it is challenging for the near future the use of coupled land surface-chemistry models (e.g.
Anav et al., 2012) which allow to account for the different feedbacks between land surfaces and
atmospheric chemistry and physics.

*Code availability.* The model used in this study is freely available and provided under the GNU
general public license 4. The source code along with the corresponding technical documentation can be
obtained from the CHIMERE web site at http://www.lmd.polytechnique.fr/chimere/. All
measurement data are publicly available
*Competing interests.* The authors declare that they have no conflict of interest.

**Acknowledgements**
We thank the investigators and the teams managing the eddy-flux sites. We also acknowledge the
entire EMEP and AIRBASE staffs for providing ground based $O_3$ data and the EMEP/MSC-W team
for anthropogenic emissions database. The computing resources and the related technical support used
for this work have been provided by CRESCO/ENEA-GRID High Performance Computing
infrastructure and its staff (http://www.cresco.enea.it). CRESCO/ENEAGRID High Performance
Computing infrastructure is funded by ENEA, the Italian National Agency for New Technologies,
Energy and Sustainable Economic Development and by National and European research programs".
Financial support was from the MITIMPACT project (INTERREG V A – Italy – France ALCOTRA).
This work was carried out within the IUFRO Task Force on Climate Change and Forest Health.








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

778 .



**FIGURES**





**Figure 1.** Comparison of hourly precipitation simulated by WRF with observations collected at four measurement sites along with changes in the vertical distribution of soil moisture ($m^3$ $m^{-3}$) during the year.






**Figure 2.** Percentage of change in the amount of O₃ removed by dry deposition over the land points (sea points are masked) computed in the time periods April-May-June (AMJ) and July-August-September (JAS). The percentage of change is defined as: [(Sim−Ref)/Ref)]*100, where Ref is the *NO_SWC* simulation and Sim represents the other simulations. A percentage of change of 25% corresponds to about 6 kg O₃ m⁻² d⁻¹.






**Figure 3.** Percentage of change in surface O$_3$ concentration (absolute values are given in Figure 4).





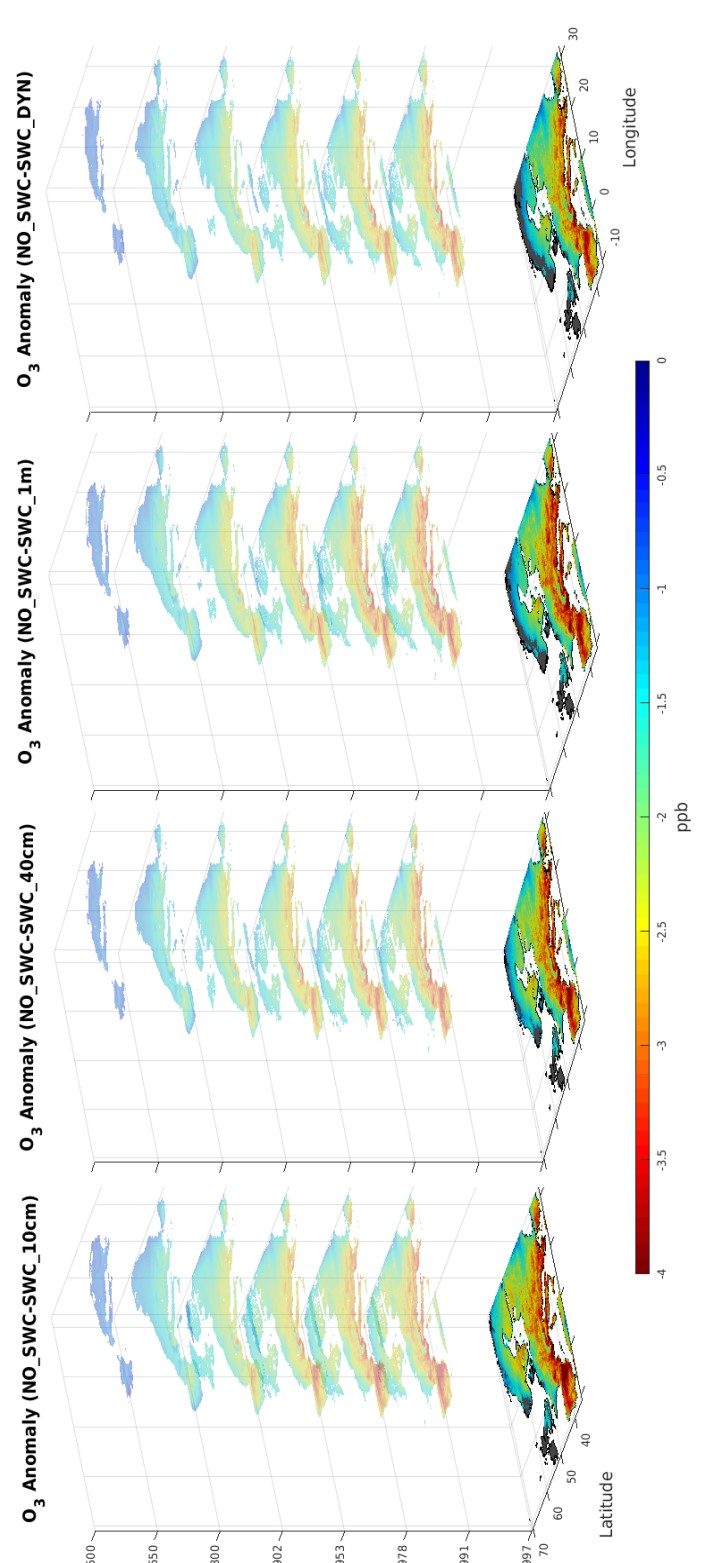

**Figure 4.** Vertical anomaly in O$_3$ concentration computed during the time period April-September.





**Figure 5.** Percentage of change in RMSE (upper panels) and correlation coefficient (lower panels) computed using hourly data in the time period April-September. The reference simulation is *NO_SWC*.


