# Peer review of "SENSITIVITY OF STOMATAL CONDUCTANCE TO SOIL MOISTURE"

_Atmospheric Chemistry and Physics, 2017_

## Short Comment (SC1) · 25 Jan 2018

My attention has been drawn to some comments in this paper about the EMEP MSC-W model (Simpson et al., 2012) and its treatment of soil water (SW) effects on ozone uptake. These comments might mislead some readers, so I would just like to clarify.

Firstly, the naming convention around EMEP and DO3SE is confusing. The DO3SE model itself is a stand-alone ozone deposition code available from the Stockholm Environment Institute at York (www.sei-international.org/do3se). The EMEP model is a 3-dimensional atmospheric chemical transport model. The deposition frameworks of the EMEP and DO3SE (then unnamed) models were developed jointly as a cooperation between several groups (EMEP, SEI-Y, FMI, University of Bradford, UK, see

[Figure]

Emberson et al., 2000a,b) in the late 1990s. Although still similar in terms of equations and parameterizsations, the EMEP and DO3SE models are very different and independent. Loosely stated though, one can say that EMEP uses the Jarvis-like stomatal conductance (gst) methods from DO3SE.

On lines 67-69 Anav et al state that 'this original formulation of the DO3SE model presented a main limitation ... ; for both forest and crops the model did not take into account the limitation due to soil water content. Lines 103-105 also suggest that SW was not accounted for in DO3SE and EMEP until the 2012 papers of Bueker et al, 2012 (DO3SE) and Simpson et al. (2012).

These comments are simply not true (as we have noted to the first author and this group before). Soil water effects have been included in the EMEP model since the 'DO3SE-like' deposition scheme was developed in the late 1990s (Emberson et al., 2000a,b). Simpson et al. (2001) and Simpson et al (2003) showed the dramatic effects that SW could have on estimated deposition velocities for a site in Portugal. Tuovinen et al (2004) also illustrated the importance of SW (using fSMD, for soil-moisture deficit) against the other Jarvis-like factors for a Portuguese meadow. These early studies showed that SW had very significant effects on modelled deposition parameters, especially in southern Europe.

Of course, estimation of SW (ether as soil moisture deficit, SMD, or soil water pressure, SWP) is extremely uncertain. Further, as noted by Tuovinen et al (2009), stomatal conductance (gst) ...

.. "is thought to be driven by SWP rather than SMD. The relationship between these two variables is both non-linear and sensitive to assumptions on soil characteristics (e.g. Jones, 1992), and even moderate uncertainties in SMD can lead to very large (orders of magnitude) uncertainties in SWP. As a result of such difficulties, a method to estimate soil water stress and its influence on gst has not, as yet, been agreed upon for EMEP or UNECE mapping purposes. However, it is clear that the influence of soil

water status on stomatal O3 flux needs to be considered in both local- and large-scale risk assessments, and especially for the Mediterranean region where soil water deficits are the norm rather than the exception. During dry summers, which may become more frequent in the future, severe drought may be experienced across the continent (Granier et al., 2007). Thus further development of methods needs to be prioritised as a matter of some urgency to ensure that flux-based risk assessments can be performed reliably at the European scale."

Thus, as these papers make clear, we have been well aware of the importance of SW effects since DO3SE-type modelling was introduced in EMEP, but we have been well aware of the considerable difficulties of both predicting and evaluating SW impacts. We have encouraged development and testing of new methods over this time period, as evidenced by the study of Bueker et al (2012).

Line 75 suggests that the role of SW has 'often been neglected in risk assessment', but this is also misleading. The main risk assessments in Europe in the context of ozone-damage are probably those of the International Cooperative Programme on Effects of Air Pollution on Natural Vegetation and Crops (ICP Vegetation, icpvegetation.ceh.ac.uk/). EMEP model results have been used extensively within ICP Vegetation, and the main focus has been to show the potential for ozone damage to well-watered sensitive vegetation. This focus is partly in recognition of the difficulties asociated with SW impacts, but also because maps of ozone-risk were indeed intended to show areas of worst-case risk. Such risk areas are valid for irrigated vegetation, or for vegetation which accesses ground-water rather than relying on precipitation - a common adaptation in dry areas.

Thus, in summary, the EMEP model (and DO3SE, they are not the same) has been capable of calculating and using SW data since the late 1990s, but we, and indeed all scientists in this field, need to be wary of relying on very uncertain predictions of SW effects when it comes to providing policy-relevant modelling results. This is clearly an important research area, and clearly a topic needing careful investigation.

References

Emberson, L., Ashmore, M., Cambridge, H., Simpson, D. & Tuovinen, J. Modelling stomatal ozone flux across Europe Environ. Poll., 2000a, 109, 403-413

Emberson, L., Simpson, D., Tuovinen, J.-P., Ashmore, M. & Cambridge, H. Towards a model of ozone deposition and stomatal uptake over Europe The Norwegian Meteorological Institute, Oslo, Norway, The Norwegian Meteorological Institute, Oslo, Norway, 2000b

Simpson, D., Tuovinen, J.-P., Emberson, L. & Ashmore, M. Characteristics of an ozone deposition module Water, Air and Soil Pollution: Focus, 2001, 1, 253-262

Simpson, D., Tuovinen, J.-P., Emberson, L. & Ashmore, M. Characteristics of an ozone deposition module II: sensitivity analysis Water, Air and Soil Pollution, 2003, 143, 123-137

Simpson, D., Benedictow, A., Berge, H., Bergström, R., Emberson, L. D., Fagerli, H., Flechard, C. R., Hayman, G. D., Gauss, M., Jonson, J. E., Jenkin, M. E., Ny\iri, A., Richter, C., Semeena, V. S., Tsyro, S., Tuovinen, J.-P., Valdebenito, Á. & Wind, P. The EMEP MSC-W chemical transport model – technical description Atmos. Chem. Physics, 2012, 12, 7825-7865

Tuovinen, J.-P., Ashmore, M., Emberson, L. & Simpson, D. Testing and improving the EMEP ozone deposition module Atmos. Environ., 2004, 38, 2373-2385

Tuovinen, J.-P., Emberson, L. & Simpson, D. Modelling ozone fluxes to forests for risk assessment: status and prospects Annals of Forest Science, 2009, 66, 401

LRTAP (2017) Chapter 3 (Mills G et al.) of the LRTAP Convention Manual of Methodologies for Modelling and Mapping Effects of Air Pollution. Available at: http:// icpvegetation.ceh.ac.uk $\sim$

---

## Referee Comment (RC1) · Anonymous Referee #1 · 4 Feb 2018

This paper describes an analysis of the seasonal effects of soil drying on ozone stomatal deposition and surface ozone concentrations. The analysis utilizes the CHIMERE chemical transport model coupled with WRF, DOS3E, and the NOAH soil models. Results show large changes in ozone deposition and surface ozone concentrations in Mediterranean climates in Europe.

My main concern is the lack of discussion. The Results section is thin and should be supplemented with quantitative information not readily derived from the maps, for example, differences resulting from the different soil moisture scenarios. Critically missing is a Discussion section, or a combined Results and Discussion, describing the reasoning, importance, and context of the results. For example, the discussion of the change in model performance is just a few sentences long and is entirely descriptive.

[Figure]

Minor comments:

The manuscript should be edited for grammar and flow. There are numerous grammatical errors.

Figure 1: Increase the font size. The titles should be changed to be more easily understood. The color bars should be labeled.

Can you add measured data to Figure 1? I understand that soil moisture measurements are made at different soil depths than the depths where the simulations are done, but they should still agree qualitatively with the gradients.

I find the model and measured precipitation correspondence difficult to discern. To my eye, it is easier to distinguish when the model and measurements do not agree. Is there some other way to represent the data? In the text, you state that the measurements are "well reproduced," but on what timescale? Weekly? Seasonally? They do not appear to coincide day-to-day.

Is there another variable that can be added to the precipitation panels that makes it visually clear why precipitation does not coincide with soil moisture seasonally?

Lines 342–346: The annual change across Europe is not a very interesting statistic. I recommend highlighting certain regions, especially the portion of Europe with a Mediterranean climate. Second, does the variability in deposition change, rather than just the mean?

Figure 2: The color scale saturates over large regions of southern Europe. I'm curious to know how large the observed percent change was.

There is little to no discussion of whether ozone deposition and ozone concentration differences were observed between soil moisture schemes. These differences, if they exist, are not apparent to me from Figures 2 and 3. Results and discussion to this point should be added.

[Figure]

I find Figure 4 and the small portion accompanying text to be unconvincing and not useful. I recommend removing this piece of the analysis.

The text concerning changes to ozone measurements and model agreement should be clarified and expanded. It isn't clear to me what the authors are communicating.

Can the authors quantitatively contextualize the change in ozone concentration results in terms of the attainment of European ozone standards?

---

## Referee Comment (RC2) · Anonymous Referee #2 · 5 Feb 2018

This study investigated the impact of soil moisture on model predicted O3 dry deposition and concentration. This is a good effort in improving current approaches handling the dry deposition process in chemical transport models as well as in studies focusing on assessing O3 impact on vegetation. By including soil moisture effect in stomatal uptake modeling, O3 dry deposition would be reduced by about 10%. While such a difference is somewhat significant, it is much smaller than the known uncertainties in most dry deposition algorithms, which is typically on the order of a factor of 2. For example, Schwede et al. (2011, A.E., 45, 1337-1346) compared one American and one Canadian models used in major monitoring networks for O3 and other gaseous species, and Flechard et al. (2011, ACP, 11, 2703-2728) compared three European and one Canadian models for nitrogen species across the NitroEurope network. Both

of these two studies suggested the differences between the commonly used dry deposition models (and thus the uncertainties in most models) being as large as a factor of 2 even on long-term average basis. In this circumstance, including soil moisture in some models may not improve the O3 prediction and may even increase the bias if the models are already biased low. This does not mean that sensitivity studies on soil moisture effects are not needed, but the existing known large bias should first be outlined, and the significance of the present study could then be elaborated. Some other specific comments are listed below.

1. Remove the introductory materials in the abstract and provide a more concise summary of the major findings.

2. Simplify the discussion of the basic concepts (especially paragraphs 3-7 in this section), and add a brief discussion on the large uncertainties in the commonly used existing schemes (as outlined above).

3. In Sections 3.2 and 3.3: where possible, first give a brief discussion on how well the original dry deposition scheme performed based on available literature so we would know if the revised version (by including soil moisture) would perform better or worse. This is important because the scientific community would depend on this finding to decide if additional effort is needed in generating soil moisture field and applying it in the dry deposition estimation.

4. In section 4, on one hand, it is stated that the dry deposition scheme is improved; and on the other hand, the bias on the model predicted O3 concentration was increased. While it is possible that the increased bias in the predicted O3 concentration was due to the large uncertainties in the other physical and chemical processes in the model, it is also possible that the original dry deposition scheme was already biased low. In the latter case, the scheme is improved in terms of including more processes, but not for the overall predicted dry deposition. Some clarifications are needed here.

---

## Author Comment (AC1) · 1 Mar 2018

AC

We thank Dr. D. Simpson for the time taken to read this manuscript and provide useful clarifications. Please find below our responses to the above comments. We are well aware of the difference between EMEP MSC-W and DO3SE model. Indeed, we would clarify that we never mention the EMEP MSC-W model in the whole paper, and we only refer to EMEP inventory as anthropogenic emissions used to drive our model. However, we agree that the reference Simpson et al (2012) at L104 could be misleading; for this reason we will remove this reference in the revised version of this manuscript. We also would highlight that our estimate of SMD is from Tuovinen et al (2004); it is based

on the Noah model which, being based on a solution of Richards' equation, estimates SWC from diffusivity, physically related to SWP, and offers the parameters needed to convert SWC into SMD. Also, Noah parameterization is much more complex than used in Tuovinen et al. (2004). These factors will much reduce the uncertainties outlined in Tuovinen et al (2009). The approximation of the SWC/SWP curve to a linear function is dictated by the Noah model itself and we hold it, within the interval (FC>SWC>WP) relevant to our study, and within the aims of our study, acceptable. We also think that, presently, Noah is the best-approximating model that can be used at continental scale. Finally, Allen et al. (1998) and Martinez-Fernández et al. (2015), both cited and discussed in the text, outlined that a linear relation between stomatal aperture and SWP is unlikely. Regarding the worst-case risk assessment, we fully agree with the above comments; in fact this part was already well discussed in the text (see L74-L78). Finally we acknowledge Dr. D Simpson for recognizing the importance of this work in assessing the role of soil moisture in atmospheric chemistry models and consequently in policy-relevant assessments.

---

## Author Comment (AC2) · 1 Mar 2018

Response to RC1:

We would like to thank the reviewer for the time taken to read and comment on this manuscript. The comments have been very helpful to improve the manuscript. We will follow your suggestions in addressing these changes in the revised version. Please find below our responses to the reviewer's comments.

RC1

This paper describes an analysis of the seasonal effects of soil drying on ozone stomatal deposition and surface ozone concentrations. The analysis utilizes the CHIMERE chemical transport model coupled with WRF, DOS3E, and the NOAH soil models. Results show large changes in ozone deposition and surface ozone concentrations in Mediterranean climates in Europe. My main concern is the lack of discussion. The Results section is thin and should be supplemented with quantitative information not readily derived from the maps, for example, differences resulting from the different soil moisture scenarios. Critically missing is a Discussion section, or a combined Results and Discussion, describing the reasoning, importance, and context of the results. For example, the discussion of the change in model performance is just a few sentences long and is entirely descriptive.

RC1

Minor comments: The manuscript should be edited for grammar and flow. There are numerous grammatical errors.

AC1

We have corrected a few typo and grammar errors in the revised manuscript.

RC1

Figure 1: Increase the font size. The titles should be changed to be more easily understood. The color bars should be labeled.

AC1

We increased the font size in Figure 1 and labeled the colorbar.

RC1

Can you add measured data to Figure 1? I understand that soil moisture measurements are made at different soil depths than the depths where the simulations are done, but they should still agree qualitatively with the gradients.

AC1

We fully agree that observed soil moisture data would help to understand whether the

model reasonably reproduces the soil water; unfortunately, over the selected sites, soil water measurement are too shallow or layers do not coincide making thus the vertical interpolation and subsequent comparison very uncertain and confusing (see for instance figure 1 at the end of the document).

RC1

I find the model and measured precipitation correspondence difficult to discern. To my eye, it is easier to distinguish when the model and measurements do not agree. Is there some other way to represent the data? In the text, you state that the measurements are "well reproduced," but on what timescale? Weekly? Seasonally? They do not appear to coincide day-to-day.

AC1

We agree that the comparison of hourly data might be confusing and not easy to read, but we also believe that only showing high frequency data allows to fully understand how the water is distributed between the different soil layers as well as evaluate the offset between rainfall events and soil water. Nevertheless, in figure 2 at the end of this document we present a more readable comparison between the simulated precipitation and the observations over the four analyzed sites. Finally, we make more clear in the revised version that rainfall events refer to the validation of hourly data.

RC1

Is there another variable that can be added to the precipitation panels that makes it visually clear why precipitation does not coincide with soil moisture seasonally?

AC1

Surely, evapotranspiration (or latent heat), runoff and snow cover would help to clarify the water dynamic into the soil; however we believe this analysis is out of the scope of this paper. In fact, the main aim here is to assess changes in atmospheric chemistry when different assumptions of water uptake by roots are used.

RC1

Lines 342–346: The annual change across Europe is not a very interesting statistic. I recommend highlighting certain regions, especially the portion of Europe with a Mediterranean climate. Second, does the variability in deposition change, rather than just the mean?

AC1

We fully agree the paper would benefit from the inclusion of a regional-based analysis; for this reason we aggregated seasonal data over climatic region derived from EEA dataset (http://discomap.eea.europa.eu/Services.aspx?agsID=9&fID=5477). This analysis allows to easily understand how mean and variability (i.e standard deviation) change between different simulations/regions/seasons. We will add figure 3 of this document in the revised paper.

RC1

Figure 2: The color scale saturates over large regions of southern Europe. I'm curious to know how large the observed percent change was.

AC1

The figure has been changes as suggested; please see the revised manuscript for further details.

RC1

There is little to no discussion of whether ozone deposition and ozone concentration differences were observed between soil moisture schemes. These differences, if they exist, are not apparent to me from Figures 2 and 3. Results and discussion to this point should be added.

AC1

Indeed, this discussion is shown in Figure 4 (and relative text in the paper); following also the next comment, we have broadened this discussion in the revised version of the manuscript.

RC1

I find Figure 4 and the small portion accompanying text to be unconvincing and not useful. I recommend removing this piece of the analysis.

AC1

We believe this figure is very useful for two reasons: 1) it clearly allows to quantify, in absolute units (i.e. not a percentage), the resulting changes in O3 concentration because of the different assumption in water uptake in the rooting zone, and 2) it shows how a process occurring within the soil affects also the concentration of gas in the upper troposphere (up to 650 hPa). According also to previous comment from reviewer, we decided to broaden this discussion in the revised paper.

RC1

The text concerning changes to ozone measurements and model agreement should be clarified and expanded. It isn't clear to me what the authors are communicating.

AC1

We have broadened this discussion in section 4 of the revised manuscript.

RC1

Can the authors quantitatively contextualize the change in ozone concentration results in terms of the attainment of European ozone standards?

AC1

We thank the reviewer for this suggestion; we have added in the revised document a new figure showing the percentage of change in the European standards used to protect vegetation and human health from ozone (i.e. AOT40 and SOMO35, respectively). Results are very interesting as we find a relevant percentage of change, reaching even the 100% in some points. More details can be found in the revised manuscript.
* * *
[Figure]

[Figure]

**Fig. 1.**

[Figure]

**Fig. 2.**

[Figure]

**Fig. 3.**

---

## Author Comment (AC3) · 1 Mar 2018

Response to RC2:

We would like to thank the reviewer for the time taken to read and comment on this manuscript and the positive comments and opinion. Please find below our responses to the reviewer's comments.

Anonymous Referee #2

This study investigated the impact of soil moisture on model predicted O3 dry deposition and concentration. This is a good effort in improving current approaches handling the dry deposition process in chemical transport models as well as in studies focusing on assessing O3 impact on vegetation. By including soil moisture effect in stomatal

uptake modeling, O3 dry deposition would be reduced by about 10%. While such a difference is somewhat significant, it is much smaller than the known uncertainties in most dry deposition algorithms, which is typically on the order of a factor of 2. For example, Schwede et al. (2011, A.E., 45, 1337-1346) compared one American and one Canadian models used in major monitoring networks for O3 and other gaseous species, and Flechard et al. (2011, ACP, 11, 2703-2728) compared three European and one Canadian models for nitrogen species across the NitroEurope network. Both of these two studies suggested the differences between the commonly used dry deposition models (and thus the uncertainties in most models) being as large as a factor of 2 even on long-term average basis. In this circumstance, including soil moisture in some models may not improve the O3 prediction and may even increase the bias if the models are already biased low. This does not mean that sensitivity studies on soil moisture effects are not needed, but the existing known large bias should first be outlined, and the significance of the present study could then be elaborated. Some other specific comments are listed below.

RC2

1. Remove the introductory materials in the abstract and provide a more concise summary of the major findings.

AC2

We shortened the introductory materials in the abstract as suggested.

RC2

2. Simplify the discussion of the basic concepts (especially paragraphs 3-7 in this section), and add a brief discussion on the large uncertainties in the commonly used existing schemes (as outlined above).

AC2

We thank the reviewer for suggesting Schwede et al. (2011) and Flechard et al. (2011)

[Figure]

papers; we have added a discussion on the uncertainties of existing dry deposition schemes in section 4 of the revised manuscript.

RC2

3. In Sections 3.2 and 3.3: where possible, first give a brief discussion on how well the original dry deposition scheme performed based on available literature so we would know if the revised version (by including soil moisture) would perform better or worse. This is important because the scientific community would depend on this finding to decide if additional effort is needed in generating soil moisture field and applying it in the dry deposition estimation.

AC2

The comparison of model's performances was already given in section 3.4, thus readers can already easily understand if the modified model would perform better or worse; additionally, in section 3.4 of the revised manuscript we have broadened the discussions comparing our results with former studies. Finally, in the last section we have broadened the discussion on the uncertainty of dry deposition, comparing this study with former publications.

RC2

4. In section 4, on one hand, it is stated that the dry deposition scheme is improved; and on the other hand, the bias on the model predicted O3 concentration was increased. While it is possible that the increased bias in the predicted O3 concentration was due to the large uncertainties in the other physical and chemical processes in the model, it is also possible that the original dry deposition scheme was already biased low. In the latter case, the scheme is improved in terms of including more processes, but not for the overall predicted dry deposition. Some clarifications are needed here.

AC2

Thanks for this suggestion, we fully agree that further clarifications are needed. For

this reason, we have broadened this discussion in section 4 adding some clarifications and references.